# BOOSTING MULTI-AGENT REINFORCEMENT LEARNING VIA TRANSITION-INFORMED REPRESENTATIONS

## ABSTRACT

Effective coordination among agents in a multi-agent system necessitates an understanding of the underlying dynamics of the environment. However, in the context of multi-agent reinforcement learning (MARL), agent partially observed information leads to a lack of consideration for agent interactions and coordination from an ego perspective under the world model, which becomes the main obstacle to improving the data efficiency of MARL methods. To address this, motivated by the success of learning a world model in RL and cognitive science, we devise a world-model-driven learning paradigm enabling agents to gain a more holistic representation of individual observation of the environment. Specifically, we present the Transition-Informed Multi-Agent Representations (TIMAR) framework, which leverages the joint transition model, i.e., the surrogate world model, to learn effective representations among agents through a self-supervised learning objective. TIMAR incorporates an auxiliary module to predict future transitions based on sequential observations and actions, allowing agents to infer the latent state of the system and consider the influences of others. Experimental evaluation of TIMAR in various MARL environments demonstrates its significantly improved performance and data efficiency compared to strong baselines such as MAPPO, HAPPO, finetuned QMIX, MAT, and MA2CL. In addition, we found TIMAR can also improve the robustness and generalization of the Transformer-based MARL algorithm such as MAT.

## 1 INTRODUCTION

Multi-agent reinforcement learning (MARL) is a rapidly growing field in the area of artificial intelligence. In recent years, significant progress has been made in the development of algorithms for MARL (Yang & Wang, 2020), and these algorithms have been applied to a wide range of tasks and environments, including game playing (Berner et al., 2019; Vinyals et al., 2019; Bellemare et al., 2013), robotics (Akkaya et al., 2019; Deitke et al., 2020; 2022), and combinatorial optimization problems (Kool et al., 2019).

Despite the many advancements made in the field of MARL, there remains a dearth of research on representation learning of the valuable information about the functionality of the world. This can lead to a lack of effective understanding of semantic information related to task goals in complex, high-dimensional scenarios, as well as a lack of analytical inferences about the states of teammates or opponents, which is crucial for efficient collaboration or competition. Relying solely on MARL algorithms may hinder the agent from acquiring such representational capabilities and make it difficult to accomplish such tasks without learning abstract representations of the world model.

Representation learning has played an important role in recent developments of single-agent reinforcement learning (RL) algorithms. In particular, self-supervised learning (SSL) has attracted increasingly more attention due to its success in both NLP and CV areas (He et al., 2020; Devlin et al., 2019; Liu et al., 2019; Lan et al., 2020). Recently, numerous works (Laskin et al., 2020; Zhu et al., 2022; Yarats et al., 2021; Schwarzer et al., 2021a; Yu et al., 2022) have borrowed insights from different areas and attempted to design auxiliary learning objectives to learn more effective representations of RL and thus improve the empirical performance. These approaches can provide the agent with a better understanding of its environment and allow the agent to learn more efficiently by focusing on the most relevant information with the help of extracted representations.

However, when meeting partially observable multi-agent systems, it is challenging to apply such self-supervision priors to learn compact and informative feature representations in MARL. One major obstacle to learning effective representations is that agents in partially observable multi-agent systems only have access to individual observations, which means that one agent's behavior influences the others' observations. As a result, building representation priors for each agent independently may fail due to imperfect and non-stationary information. In other words, it is challenging to learn representations that can provide a more holistic observation of the environment and serve as valuable supervision to explicitly guide the model learning how to collaborate among agents.

We tackle this challenge by designing an approach to enhance the data efficiency of MARL algorithms to learn valuable information about the functionality of the environment's world model through an SSL way in the latent space. As shown in Figure 1, our insight is that humans acquire a substantial amount of background knowledge about the world through passive observation. Scholars have hypothesized that this common-sense information plays a crucial role in enabling intelligent behavior, including the sample-efficient acquisition of new concepts (Sarkar & Etemad, 2020), grounding (Assran et al., 2023), and planning (LeCun, 2022). As a result, with the help of implicit inference with a virtual ego world

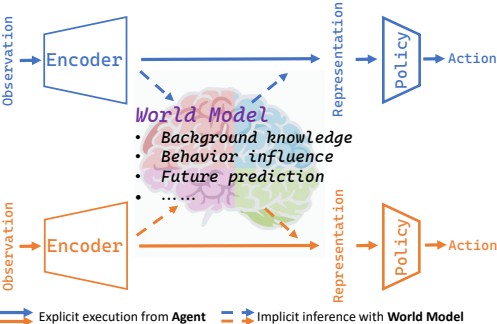

Figure 1: The insight for improving MARL methods with the world model.

model, an agent can obtain better information, such as background knowledge, behavior influence, future prediction, etc., for its explicit execution in the environment.

In this work, we propose a novel representation learning framework that suits MARL, named **T**ransition-**I**nformed **M**ulti-**A**gent **R**epresentations, dubbed TIMAR, to improve data efficiency and performance of MARL further. The idea behind TIMAR is to ground representation among agents with the joint transition model, i.e. the surrogate world model. In addition to the encoder in previous MARL approaches, we introduce an auxiliary Transformer-liked module (Vaswani et al., 2017) to model the interaction among agents. Specifically, we first treat latent representations of local observations of all agents as the sequence of masked contexts of the global state. Then we combine the sequential observation representations and action embeddings to let the Transformer module inform the observation representation of the next timestep. Inspired by the success of self-supervised learning objectives in efficient RL (Laskin et al., 2020; Schwarzer et al., 2021a), we adopt BYOL's (Grill et al., 2020) loss to train the original encoder and Transformer jointly, meanwhile ensuring the consistency between the informed transitions and the ground truth.

To evaluate our proposed algorithm, we try our framework on strong MARL algorithms and construct extensive experiments on multiple common-used cooperative MARL benchmarks, including both vision- and state-based environments in discrete and continuous scenarios (Samvelyan et al., 2019; Panerati et al., 2021; de Witt et al., 2020). We compare our approach against current state-of-the-art baselines such as finetuned QMIX (Hu et al., 2021), HAPPO (Kuba et al., 2022), Multi-Agent Transformer (Wen et al., 2022), and MA2CL (Song et al., 2023). The results demonstrate its superior performance and data efficiency in these environments, meaning that TIMAR can learn more impactive representations from our designed joint-transition-model-based self-supervised learning paradigm compared with baselines. In addition, we show that TIMAR can also improve the robustness and generalization of the Transformer-based MARL algorithm such as MAT.

## 2 RELATED WORK

### 2.1 OVERVIEW OF SELF-SUPERVISED LEARNING

Self-supervised learning empowers us to exploit a variety of labels that come with the data for free. With self-supervised learning, we can utilize inexpensive unlabeled data and establish the learning objectives properly from designed pretexts to gain supervision from the data itself. SSL has been developed in CV and NLP areas and can be divided into the various self-supervised pretexts in the literature into four broad families (Ericsson et al., 2022): *Masked Prediction*, *Transformation Prediction*, *Instance Discrimination*, and *Clustering*. (1) Masked Prediction methods (Mikolov et al.,

2013; Baevski et al., 2020; Pathak et al., 2016; Hu et al., 2020) mask a portion of word tokens or image pixels from the input sentence or image and train the model to predict the masked components to obtain effective representations. (2) Transformation Prediction methods (Gidaris et al., 2018; Sarkar & Etemad, 2020; Xu et al., 2019) apply a transformation that maps from canonical views to alternative views and trains the model to predict what transformation has been applied. (3) Instance Discrimination methods (Velickovic et al., 2019; Chen et al., 2020; He et al., 2020; Tian et al., 2020) apply some transformation process in one instance to obtain multiple views of it and attempt to formalize the contrastive instance discrimination. (4) And Clustering methods (Caron et al., 2018; 2020; Zhan et al., 2020; Alwassel et al., 2020) focus on dividing the training data into several groups with high intra-group similarity and low inter-group similarity. We recommend readers read Ericsson et al. (2022) to get more information.

## 2.2 Self-Supervised Learning for RL

There exist substantial works taking advantage of SSL techniques to promote representation learning in RL. A popular approach is to jointly learn policy learning objectives and auxiliary objectives. As for constructing auxiliary SSL objectives, the primary way is to build multiple views of the same input through masked-latent reconstruction or dynamic models with augmentations. For instance, Laskin et al. (2020) and Zhu et al. (2022) attempt to extract high-level features from raw pixels using contrastive learning and perform off-policy control on top of the extracted features. Other works (Schwarzer et al., 2021a; Yu et al., 2021b; 2022; Zhang et al., 2021) leverage a dynamic model to obtain a predicted version of the subsequent observation and then use contrastive learning to enforce consistency between the raw future observation and the prediction version of it in latent space. Another alternative way of obtaining good representations is to pre-train the observation encoder to learn effective representations before policy learning (Yarats et al., 2021; Stooke et al., 2021; Schwarzer et al., 2021b; Yang & Nachum, 2021; Campos et al., 2021).

## 2.3 Self-Supervised Learning for MARL

As far as we know, only a few works (Shang et al., 2021; Zhang et al., 2022; Song et al., 2023; Guan et al., 2022; Lin et al., 2021) consider promoting representation in MARL. Shang et al. (2021) task each agent to predict its future location, arriving at an agent-centric predictive objective to be combined in their proposed agent-centric attention module in the football game. Zhang et al. (2022) is a model-based MARL method that proposed a graph-assisted predictive state representation learning framework that leverages the agent connectivity graphs to aggregate local representations computed by each agent. Guan et al. (2022) designs a permutation invariant message encoder to generate common information-aggregated representation from messages and optimize it via reconstructing and shooting future information in a self-supervised manner. And Lin et al. (2021) formulates communication grounding as a representation learning problem and proposes to use observation autoencoding to learn a common grounding across all agents. Note that the SSL prior proposed in Shang et al. (2021) only be used in football-like environments and is not flexible. Additionally, our method aims to build a general plugin for model-free MARL approaches so that model-based and communication-based MARL methods are not directly comparable to our method.

We focus on the auxiliary-task-based studies in this work. The most similar work is Song et al. (2023), which encourages learning representation to be both temporal and agent-level predictive by reconstructing the masked agent observation in latent space. Specifically, it uses an attention reconstruction model for recovering and the model is trained via contrastive learning. Different from Song et al. (2023), our method leverages the joint-embedding predictive architecture to learn the surrogate multi-agent world to capture effective knowledge for better multi-agent decision-making.

## 3 Our Method

### 3.1 Preliminaries and Background

**Problem formulation:** Cooperative MARL problems are often modeled by decentralized Partially Observable Markov Decision Processes (Dec-POMDPs, Oliehoek & Amato (2016)) $(\mathcal{N}, \mathcal{S}, \{\mathcal{A}_i\}, \mathcal{T}, R, \Omega, \mathcal{O}, \gamma)$. Here, $\mathcal{N} = 1, \ldots, n$ is the set of agents, $\mathcal{S}$ is the set of states,

$\mathcal{A} = \times_i \mathcal{A}_i$ is the set of joint actions, $\mathcal{T}$ is a set of conditional transition probabilities between states, $\mathcal{T}(s, \boldsymbol{a}, s') = P(s' \mid s, \boldsymbol{a})$, $R : \mathcal{S} \times \mathcal{A} \to \mathbb{R}$ is the reward function, $\mathcal{O} = \times_i \mathcal{O}_i$ is a set of observations for agent $i$, $\Omega$ is a set of conditional observation probabilities $\Omega(s', \boldsymbol{a}, \boldsymbol{o}) = P(\boldsymbol{o} \mid s', \boldsymbol{a})$, and $\gamma \in [0, 1]$ is the discount factor. At each time step, each agent selects an action $a_i$, and the state updates according to the transition function (using the current state and the joint action). Then each agent receives its observation based on the observation function $\Omega(s', \boldsymbol{a}, \boldsymbol{o})$ (using the next state and the joint action) and a reward is generated for the entire team according to the reward function $R(s, \boldsymbol{a})$. The goal is to maximize the expected cumulative reward over a finite or infinite time horizon.

**MARL algorithms:** In deep MARL, we use neural networks to process joint observations and make decisions. A common-used paradigm is *centralized training for decentralized execution* (CTDE), which allows agents to access global information and opponents' actions during the training phase and use individual observation only in the inference phase. In CTDE approaches (Lowe et al., 2017; Yu et al., 2021a; Kuba et al., 2022; Rashid et al., 2018; Wang et al., 2021), observation representations are generated from the encoder of the decentralized part of the algorithm, e.g. the actor in policy gradient-based methods and the backbone in value-based methods. Let $f_\theta$ denote the encoder parameterized by $\theta$, that is $\hat{\boldsymbol{o}}_t^i = f_\theta(\boldsymbol{o}_t^i)$. Another powerful MARL approach is Multi-Agent Transformer (MAT). It is a Transformer-liked architecture and takes the joint observations as input to obtain the representations. In MAT, the transformation process for observations into representations can be described as $\hat{\boldsymbol{o}}_t^{i_{1:n}} = f_\theta(\boldsymbol{o}_t^{i_{1:n}})$, where $i_{1:n}$ denotes an arbitrary order for agents. We denote all other parts of the MARL methods as $f_\phi$ parameterized by $\phi$, including the value head or the policy head. Different MARL algorithms use one or two of these heads and feed the representations to calculate the value loss or policy loss in the MARL branch.

## 3.2 TRANSITION-INFORMED MULTI-AGENT REPRESENTATIONS

Transition-Informed Multi-Agent Representations (TIMAR) is an auxiliary objective to promote representation learning in MARL. The core idea of TIMAR is to take advantage of a world-model-driven SSL approach to promote representation learning in MARL, toward addressing the challenge of imperfect and non-stationary observations. To achieve the goal, an intuitive way is to leverage a surrogate world model to inform the joint transition of the next timestep so that we can obtain a different view of the ground-truth next-timestep observations sampled from the replay buffer. As a result, executing consistency across different views of observations can lead to better representations generated from encoder networks. Furthermore, the core process in the joint transition model of TIMAR is to **implicitly reconstruct the global state and then infer the future observation representation of each agent**. This enables the better use of agent-cross information when learning observation and action representations, further enhancing the understanding of MARL agents for individual messages. We will introduce the components shown in the framework in the following subsections.

**Framework overview.** In TIMAR, as shown in the left part of Figure 2, in the training phase, a stack of $K + 1$ consecutive $n$-agent joint observations $\boldsymbol{o}_{t:t+K}^{i_{1:n}}$ is first sampled from the replay buffer. Then we encode the oldest timestep observations $\boldsymbol{o}_t^{i_{1:n}}$ with MARL algorithm's encoder to get the joint-observation representations. Apart from using them in the MARL optimization branch to train the whole online networks, the $t$-th timestep representations will also be feedforward into the transition model with the action embedding sequence to predict future observation representations. After repeating the prediction process $K$ times, we can obtain $K$ joint representations, i.e. $\tilde{\boldsymbol{o}}_{t+1:t+K}$. Meanwhile, we take the rest of the joint observations $\boldsymbol{o}_{t+1:t+K}$ into the momentum encoder to generate the ground-truth version of the $K$-timesteps observation representations. Finally, we use these two views to calculate the SSL-style transition-informed loss to encourage effective representations in both temporal and agent-level dimensions. The processes of *encoding, transition model* and *transition-informed loss* are introduced in detail below.

**(i) Encoding observations and actions:** Given a specific MARL algorithm, We use its encoder $\theta$ as the *online* observation encoder to transform the joint observations into representations. Concretely, for MAT, taking the observation sequence of arbitrary order $\boldsymbol{o}_t$ as input, the online observation encoder applies a self-attention mechanism and obtains post-interaction representations of agents, as $\hat{\boldsymbol{o}}_t$. Similarly to the online observation encoder, the online action encoder accepts both the origin

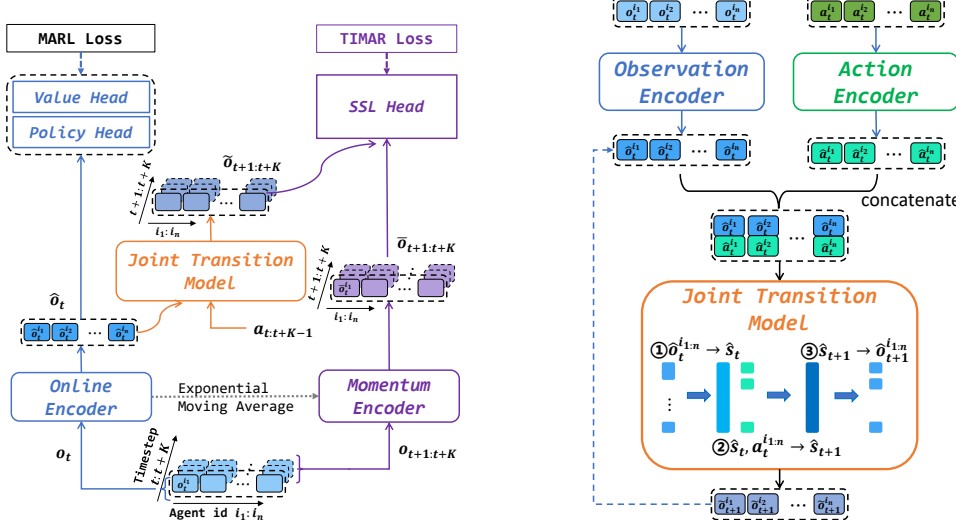

Figure 2: Demonstrations of the TIMAR. *Left*: illustration of the full TIMAR method. *Right*: illustration of the joint transition model.

action sequence $\boldsymbol{a}_t$ and observation representations $\boldsymbol{o}_t^{i_{1:n}}$ and the output action representations $\hat{\boldsymbol{a}}_t$ through the cross-attention mechanism. In contrast, CTDE methods process individual observations in parallel to obtain representations. Besides, we use a separate action encoder to transform the actions into action embeddings. We employ these representations with a goal that motivates them to forecast future observation representations up to a given temporal offset $K$, iteratively. Following prior work Schwarzer et al. (2021a); Zhu et al. (2022); Yu et al. (2021b; 2022), we utilize another observation encoder to encode original observations. This *target* encoder has the same architecture as the online observation encoder, and its parameters are an exponential moving average (EMA) of the online observation encoder parameters. Denoting the target observation encoder as $\bar{\theta}$ and the momentum coefficient as $\tau \in [0, 1)$, the update scheme of the target observation encoder is:

$$\bar{\theta} \leftarrow \tau\bar{\theta} + (1 - \tau)\theta. \tag{1}$$

**(ii) Joint Transition Model.** We construct the forecasting version of future observation representations using a transformer-based joint transition model $\hat{\mathcal{T}}$. In other words, we *treat the individual observations as a sequence of masked contexts of global state* in the joint transition model. The architecture of the Transformer encoder is leveraged in the joint transition model, and (a) contains $L$ Multi-Head Self-Attention (MHSA) layers without masks, and (b) takes the sequence of concatenated observations with action representations as input tokens and then outputs the sequence of observation representations of the subsequent timestep. We obtain the $t$-th observation and action representations by feeding the origin observation sequence and action sequence into the online and action encoder, as mentioned above. The input tokens of the latent joint transition model can be mathematically represented as:

$$\boldsymbol{x} = [\hat{o}_t^{i_1}||\hat{a}_t^{i_1}, \ldots, \hat{o}_t^{i_n}||\hat{a}_t^{i_n}], \tag{2}$$

where $||$ denotes concatenation operator.

For any $l \in [L]$, the process of passing the token sequence through the $l$-th layer of the joint transition model can be mathematically described as follows:

$$\begin{aligned}
\boldsymbol{h}^l &= \text{MHSA}\left(\text{LN}\left(\boldsymbol{x}^l\right)\right) + \boldsymbol{x}^l, \\
\boldsymbol{x}^{l+1} &= \text{FFN}\left(\text{LN}\left(\boldsymbol{h}^l\right)\right) + \boldsymbol{h}^l.
\end{aligned} \tag{3}$$

Here, LN and FFN denote the LayerNorm and the Feed-Forward Network mentioned in Vaswani et al. (2017). Note that if the permutation order is known, one can also add agent ids' embedding and positional embedding on $\boldsymbol{x}$. And we only select the odd elements of the output tokens of the joint transition model as the corresponding predictive results for the latent future representations inferred from previous observation and action representations.

Furthermore, in the $k$-th step of generating future representations where $k = 2, \ldots, K$, we use internal representations, i.e., generated from the joint transition model, instead of the online observation

encoders as the input latent observation tokens. The process mentioned above can be denoted as

$$\tilde{\boldsymbol{o}}_{t+1} = \hat{\mathcal{T}}\left(\hat{\boldsymbol{o}}_t, \boldsymbol{a}_t\right),$$
$$\tilde{\boldsymbol{o}}_{t+k} = \hat{\mathcal{T}}\left(\tilde{\boldsymbol{o}}_{t+k-1}, \boldsymbol{a}_{t+k-1}\right), \quad \forall k = 2, \ldots, K. \tag{4}$$

It is worth noting that both the joint transition model and the calculating process of the transition-informed loss operate in the latent space, thus avoiding pixel-based reconstruction objectives and making TIMAR robust for vision-based and state-based MARL settings.

Based on the description of the process of the joint transition model, one can see that the module first reconstructs the global state from individual observations and then predicts the future state of the next timestep. Finally, it implements the observation mapping functions for each agent. In this way, the joint transition model must infer the influences caused by others and try to integrate all the imperfect information. As a result, executing consistency across different views of individual observations can lead to better representations generated from encoder networks. The illustration of the joint transition model is shown in the right part of Figure 2.

**(iii) Transition-informed loss.** Motivated by the success of BYOL Grill et al. (2020) in SSL and sample-efficient RL Schwarzer et al. (2021a); Yu et al. (2021b; 2022), we compute the future prediction loss of TIMAR by calculating the cosine similarities between the predicted and observed representations. Concretely, from the outputs of the joint transition model, i.e. the sequence of observation representations set $\tilde{\boldsymbol{o}}_{t+1:t+K}$, we use a projection head $g$ and a prediction head $q$ to obtain the final sequence of predictions result in $\tilde{\boldsymbol{y}}_{t+1:t+K} = q(g(\tilde{\boldsymbol{o}}_{t+1:t+K}))$. Then we utilize a target projection head $\bar{g}$ (i.e. follows the same EMA update strategy in the target observation encoder) to process the encoded results of original observations, which is denoted as $\bar{\boldsymbol{y}}_{t+1:t+K} = \bar{g}(\bar{\boldsymbol{o}}_{t+1:t+K})$ where $\bar{\boldsymbol{o}}_{t+1:t+K} = \bar{\theta}(\boldsymbol{o}_{t+1:t+K})$. Here, we apply a stop-gradient operation as illustrated in Figure 3 to avoid model collapse, following BYOL. Finally, TIMAR's objective is to enforce the final prediction result in $\tilde{\boldsymbol{y}}_{t+1:t+K}$ to be as close to its corresponding target $\bar{\boldsymbol{y}}_{t+1:t+K}$. And we construct the following cosine similarities between the normalized predictions and the target projections overall agents and the offset timesteps:

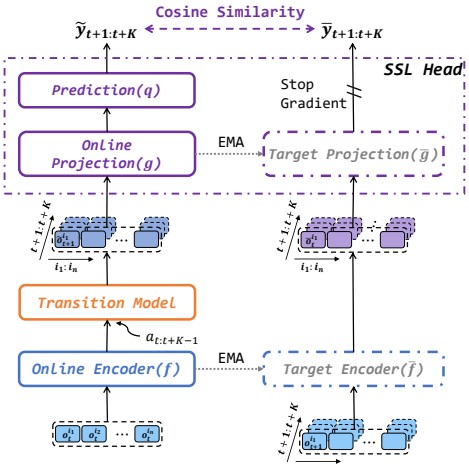

Figure 3: Loss calculation for TIMAR.

$$\mathcal{L}_{\text{TIMAR}} = -\frac{1}{Kn} \sum_{k=1}^{K} \sum_{i=1}^{n} \left(\frac{\tilde{y}_{t+k}^i}{\left\|\tilde{y}_{t+k}^i\right\|_2}\right)^{\top} \left(\frac{\bar{y}_{t+k}^i}{\left\|\bar{y}_{t+k}^i\right\|_2}\right) \tag{5}$$

**Total learning objective:** The proposed TIMAR is an auxiliary task that is optimized in conjunction with MARL. Therefore, the overall loss function is:

$$\mathcal{L}_{total} = \mathcal{L}_{\text{MARL}} + \lambda \mathcal{L}_{\text{TIMAR}} \tag{6}$$

where $\mathcal{L}_{\text{MARL}}$ and $\mathcal{L}_{\text{TIMAR}}$ are the MARL loss and our proposed transition-informed representation learning objective, respectively. $\lambda$ is a hyperparameter for balancing the items. It is worth noting that, unlike other suggested SSL algorithms in CV and RL, TIMAR can be employed with or without data augmentation, especially in situations where data augmentation is unavailable or counterproductive. Moreover, TIMAR mainly focuses on capturing the relationships among agents via the joint transition model. The proposed framework can also be transferred to other MARL algorithms that follow the centralized training decentralized execution (CTDE) paradigm, such as MAPPOYu et al. (2021a)/HAPPO Kuba et al. (2022) and QMIX Rashid et al. (2018)/QPLEX Wang et al. (2021), etc.

### 3.3 IMPLEMENT DETAILS FOR TIMAR

In practice, we implement instantiations of TIMAR on the basis of the recently proposed state-of-the-art method MAT and the commonly used CTDE method, QMIX. On one hand, we apply TIMAR

only upon the encoder of MAT, which contains an MLP-based embedding layer for original inputs and a one-layer transformer encoder for agent-level information interaction. On the other hand, as for QMIX and other CTDE-liked MARL methods, we use sequential layers before the RNN units in the network as TIMAR's online encoder.

Besides, we sample a unique batch of $B'$ samples from the trajectories collected using the latest policy, both for on-policy MAT and off-policy QMIX. For the projection and prediction head, we do not use BatchNorm Layer and replace ReLU with GELU activation units, which is different from what BYOL does. As for vision-based settings, we use three convolutional layers with a ReLU layer after each convolutional layer, which is the same as DQN's, as the feature extractor in all algorithms.

Finally, our code is based on MAT and finetuned QMIX's official codebase and the full hyperparameters of TIMAR can be found in Appendix C.

## 4 EXPERIMENTS

In this section, we consider a series of MARL benchmarks to evaluate TIMAR, including Multi-agent MuJoCo (MA-MuJoCo), the StarCraftII Multi-agent Challenge (SMAC), and Multi-Agent Quadcopter Control (MAQC). The result demonstrates that TIMAR achieves performance and efficiency superior to those of strong MARL baselines, including the mentioned work MA2CL (Song et al., 2023). We also take an analysis of the reason for TIMAR's effectiveness. Moreover, the extended result shows that TIMAR can also improve the robustness and generalization of the sequential-modeling-based MARL algorithm.

### 4.1 PERFORMANCE AND EFFICIENCY

#### 4.1.1 MULTI-AGENT MUJOCO

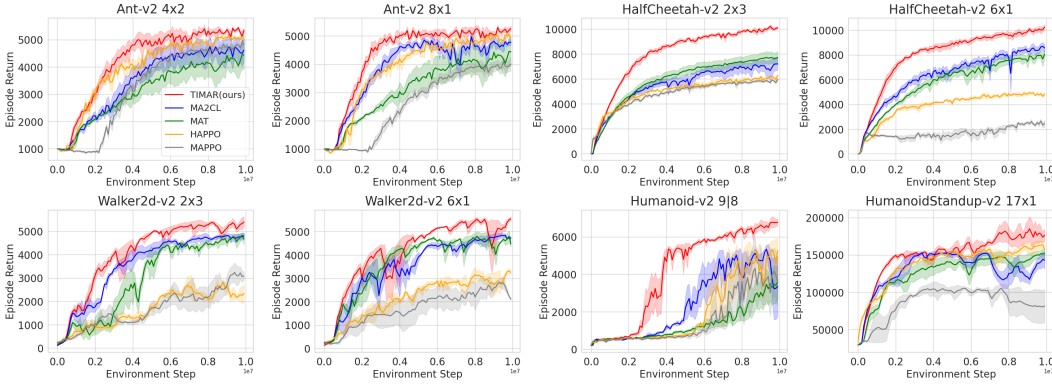

Figure 4: Comparisons of average episode return of compared algorithms on Multi-Agent MuJoCo. TIMAR consistently outperforms MA2CL, refreshing the SOTA results for on-policy algorithms.

MA MuJoCo (de Witt et al., 2020) is a common-used benchmark for continuous cooperative multi-agent robotic control. Starting from the popular single-agent robotic MuJoCo (Todorov et al., 2012) control suite included with OpenAI Gym Brockman et al. (2016), it creates a wide variety of novel scenarios in which multiple agents within a single robot have to solve a task cooperatively.

Since its heterogeneous-agent setting and the advantages of approaches with sequential updating scheme shown in recent studies (Kuba et al., 2022; Wen et al., 2022; Zhong et al., 2023), we try our method on one of the state-of-the-art (SOTA) MARL algorithm MAT and evaluate it on predefined tasks in MA MuJoCo and select MAPPO, HAPPO, MAT, and MA2CL as compared baseline.

### 4.1.2 THE STARCRAFT MULTI-AGENT CHALLENGE (SMAC)

The StarCraft Multi-Agent Challenge (Samvelyan et al., 2019), briefly called SMAC, is a benchmark environment for training and evaluating multi-agent reinforcement learning (MARL) algorithms. It is based on the popular real-time strategy game StarCraft II and provides a challenging testbed for MARL research due to the complexity of the game and the need for agents to coordinate and compete with each other. The SMAC environment is open-source and widely used in the research community, making it a common benchmark for evaluating the performance of MARL algorithms.

Different from the settings of MAQC and MA MuJoCo, we select QMIX (Rashid et al., 2018; Hu et al., 2021) as the basic algorithm for incorporating our method since it is almost the most common-used CTDE method in the SMAC domain. This will also demonstrate TIMAR's generalization for CTDE methods and Value-Decomposition-based paradigms in the MARL area.

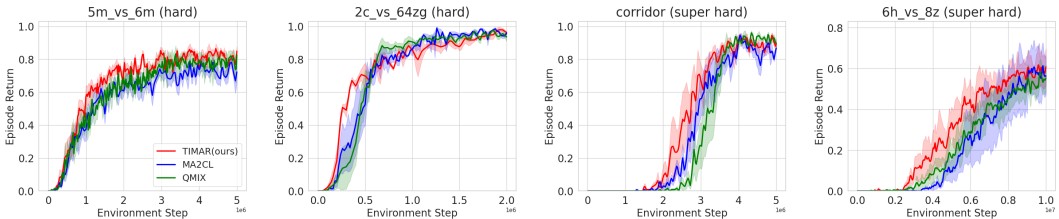

Figure 5: Comparisons of the winning rate of TIMAR against MA2CL and finetuned QMIX in SMAC testbed.

### 4.1.3 MULTI-AGENT QUADCOPTER CONTROL

To evaluate whether our proposed TIMAR is powerful in vision-based MARL settings, we run it on three physics-based cooperative tasks in Multi-Agent Quadcopter Control (MAQC) (Panerati et al., 2021). MAQC is an open-source, OpenAI Gym-like multi-quadcopter simulator that provides vision-based observations and multi-agent controlling interfaces. Observations include video frames from the perspective of each drone (toward the positive direction of the local x-axis) for the RGB ($\in \mathbb{R}^{48 \times 48 \times 4}$), depth, and segmentation ($\in \mathbb{R}^{48 \times 48 \times 1}$) views. The action of drones is continuous velocity and the magnitude of the velocity. We recommend readers get more information about the descriptions of MAQC in Appendix A.3. We test TIMAR, MA2CL, MAT, HAPPO, and

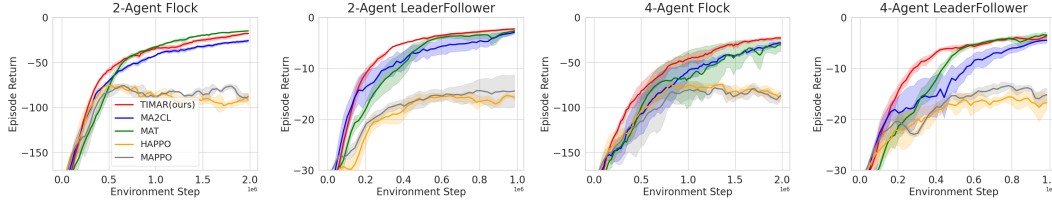

Figure 6: Comparisons of the episode return of TIMAR against MA2CL, MAT, HAPPO, and MAPPO in Multi-Agent Quadcopter Control environment.

MAPPO at 4 subtasks in MAQC, which contains two fly-controlling scenarios (named *Flock* and *LeaderFollower*) with two and four agents, respectively. The results shown in Figure 6 demonstrate TIMAR can improve data efficiency of MAT better than MA2CL for visual signals.

### 4.2 ANALYSIS ABOUT WHY TIMAR WORKS

In this part, we attempt to understand how TIMAR improves the augmented MARL approaches. Since the encoder in MAT and QMIX is the backbone of the value estimation branch in the whole algorithm, we plot the training curve of both TIMAR, MA2CL, and corresponding MARL methods

in four scenarios of MA MuJoCo and SMAC. Results are shown in Figure 7. We would like to posit whether the global value function approximation in MAT or Q values for taken actions in QMIX would be enhanced from the compact representation built upon TIMAR. One can see that TIMAR's value loss is lower than MAT's and the Q value is higher than QMIX's respectively. The more accurately the value function fits the better the policy optimization effect.

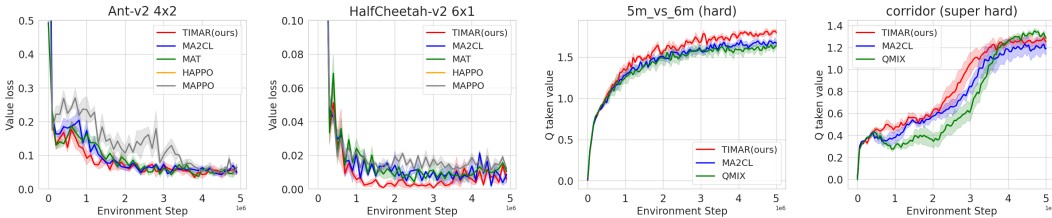

Figure 7: An illustration of the TIMAR's benefit in MAT and QMIX.

## 4.3 GENERALIZATION AND ROBUSTNESS

Since Transformer-based models often demonstrate strong performance on generalization and robustness, we believe that TIMAR can also improve MAT's corresponding abilities. And we design two experiments to validate such an assumption on HalfCheetah 6x1 of MA MuJoCo: one is evaluating the performance for different disabled joints on the training process to check the robustness of TIMAR, and another is validating TIMAR's performance for different partial observable situations for the same task to evaluate its generalization. We list the results

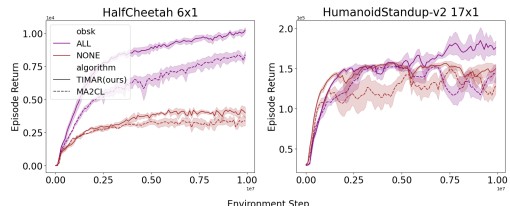

Figure 8: Generalization evaluation results for different observable views (i.e. *obsk*) in the two scenarios of MA MuJoCo.

in Figure 8 and Figure 9, which tell us that TIMAR can not only boost the original sample efficiency and performance of Transformer-based MAT but also can further improve its generalization and robustness with the learning objective of the world model.

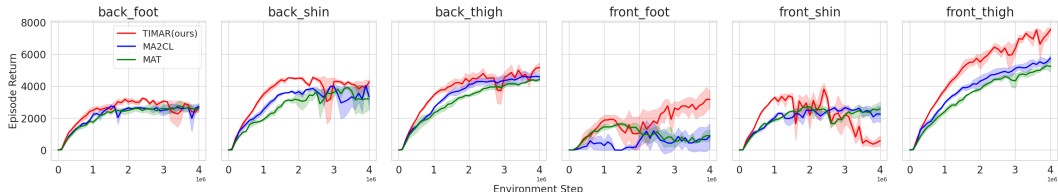

Figure 9: Robustness evaluation on HalfCheetah 6x1 of MA MuJoCo with different disabled joints.

## 5 CONCLUSION

In this paper, we introduce the Transition-Informed Multi-Agent Representations (TIMAR), a self-supervised representation learning objective designed to improve the data efficiency of MARL algorithms with the help of the joint transition, i.e. the surrogate world model. TIMAR treats the individual observations as a masked sequence and learns the impactive representations that are jointly temporally predictive and consistent across different views overall agents, by implicitly reconstructing the global state and directly predicting representations of observations produced by a joint transition model and a target encoder. Experimental results on both vision-based and state-based cooperative MARL benchmarks (i.e. MAQC, MA MuJoCo, and SMAC) demonstrate that TIMAR can further improve data efficiency and performance for used MARL backbone algorithms such as QMIX, MAT, and MA2CL. Besides, TIMAR can also bring benefits for MAT's generalization and robustness.

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

## A  EXTENDED BACKGROUND

### A.1  THE TRANSFORMER MODEL AND THE ATTENTION MECHANISM

Transformer Vaswani et al. (2017) was created originally for machine translation jobs (e.g., input English, output French). It has an encoder-decoder structure in which the encoder maps an input sequence of tokens to latent representations and then the decoder generates a sequence of desired outputs in an auto-regressive manner, with the Transformer taking all previously generated tokens as input at each step of inference. The scaled dot-product attention, which captures the interrelationship of input sequences, is a critical component of the Transformer. The attention function is written as Attention $(\mathbf{Q}, \mathbf{K}, \mathbf{V}) = \mathrm{softmax}\left(\frac{\mathbf{Q}\mathbf{K}^T}{\sqrt{d_k}}\right)\mathbf{V}$, where the $\mathbf{Q}, \mathbf{K}, \mathbf{V}$ corresponds to the vector of queries, keys and values, which can be learned during training, and the $d_k$ represent the dimension of $\mathbf{Q}$ and $\mathbf{K}$. Self-attentions refer to cases when $\mathbf{Q}, \mathbf{K}, \mathbf{V}$ share the same set of parameters.

### A.2  EXISTING METHODS IN MARL

We will now give a quick overview of common-used MARL algorithms.

**QMIX** (Rashid et al., 2018; Hu et al., 2021) can be thought of as an extension of DQN to the Dec-POMDP setting. The joint optimal action is found by forcing the joint $Q$ to adhere to the individual global max (IGM) principle(Son et al., 2019), which states that the joint action can be found by maximizing individual agents' $Q_i$ functions:

$$\arg\max_{\boldsymbol{a}} Q(s, \boldsymbol{\tau}, \boldsymbol{a}) = \begin{cases} \arg\max_a Q_1(\tau_1, a_1) \\ \arg\max_a Q_2(\tau_2, a_2) \\ \cdots \\ \arg\max_a Q_n(\tau_n, a_n) \end{cases} \tag{7}$$

This central $Q$ is trained to regress to a target $r + \gamma \hat{Q}(s, \boldsymbol{\tau}, \boldsymbol{a})$ where $\hat{Q}$ is a target network that is updated slowly. The central $Q$ estimate is computed by a mixing network, whose weights are conditioned on the state, which takes as input the utility function $Q_i$ of the agents. The weights of the mixing network are restricted to be positive, which enforces the IGM principle(Son et al., 2019) by ensuring the central $Q$ is monotonic in each $Q_i$.

**MAPPO** Yu et al. (2021a) was the first and most straightforward technique for implementing PPO in MARL. It provides all agents with the same set of parameters and updates the shared policy based on the aggregated trajectories of the agents. In detail, at iteration $k + 1$, it optimizes the policy parameter $\theta_{k+1}$ by maximizing the clip objective of

$$\sum_{i=1}^n \mathbb{E}_{\mathbf{o} \sim \rho_{\boldsymbol{\pi}_{\theta_k}}, \mathbf{a} \sim \boldsymbol{\pi}_{\theta_k}} \left[ \min\left( \frac{\pi_\theta\left(\mathrm{a}^i \mid \mathbf{o}\right)}{\pi_{\theta_k}\left(\mathrm{a}^i \mid \mathbf{o}\right)} A_{\boldsymbol{\pi}_{\theta_k}}(\mathbf{o}, \mathbf{a}), \mathrm{clip}\left( \frac{\pi_\theta\left(\mathrm{a}^i \mid \mathbf{o}\right)}{\pi_{\theta_k}\left(\mathrm{a}^i \mid \mathbf{o}\right)}, 1 \pm \epsilon \right) A_{\boldsymbol{\pi}_{\theta_k}}(\mathbf{o}, \mathbf{a}) \right) \right],$$

where the clip operator (if required) trims the input value to keep it inside the interval $[1 - \epsilon, 1 + \epsilon]$. Enforcing parameter sharing, on the other hand, is analogous to imposing a restriction $\theta^i = \theta^j, \forall i, j \in \mathcal{N}$ on the joint policy space, which might result in a suboptimal conclusion that is exponentially worse Kuba et al. (2022). This encourages the development of more principled heterogeneous-agent trust-region approaches, such as HAPPO.

**HAPPO** is one of the SOTA algorithms that completely exploits Multi-Agent Advantage Decomposition Kuba et al. (2021) to provide multi-agent trust-region learning with monotonic improvement guarantees. During an update, the agents choose a permutation $i_{1:n}$ at random, and then, in the sequence in which the permutation was chosen, each agent $i_m$ picks $\pi_{\mathrm{new}}^{i_m} = \pi^{i_m}$ that maximizes the aim of

$$\mathbb{E}_{\mathbf{o} \sim \rho_{\boldsymbol{\pi}_{\mathrm{old}}}, \mathbf{a}_{1:m-1}^{i_1} \sim \pi_{\mathrm{new}}^{i_{1:m-1}}, \mathrm{a}^i m \sim \pi_{\mathrm{old}}^{i_m}}^{i_m} \left[ \min\left( \mathrm{r}\left(\pi^{i_m}\right) A_{\boldsymbol{\pi}_{\mathrm{old}}}^{i_{1:m}}\left(\boldsymbol{o}, \mathbf{a}^{i_{1:m}}\right), \mathrm{clip}\left( \mathrm{r}\left(\pi^{i_m}\right), 1 \pm \epsilon \right) A_{\boldsymbol{\pi}_{\mathrm{old}}}^{i_{1:m}}\left(\boldsymbol{o}, \mathbf{a}^{i_{1:m}}\right) \right) \right],$$

where $\mathrm{r}\left(\pi^{i_m}\right) = \pi^{i_m}\left(\mathrm{a}^{i_m} \mid \mathbf{o}\right) / \pi_{\mathrm{old}}^{i_m}\left(\mathrm{a}^{i_m} \mid \mathbf{o}\right)$. It is worth noting that the expectation is placed over the newly-updated prior agents' policies, i.e., $\pi_{\mathrm{new}}^{i_{1:m-1}}$; this reflects an intuitive understanding

that, according to Theorem (1), the agent $i_m$ responds to its preceding agents $i_{1:m-1}$. However, one disadvantage of HAPPO is that agent policies must adhere to the sequential updating strategy in the permutation, preventing it from being executed in parallel.

**Multi-Agent Transformer** effectively casts cooperative MARL into Sequential Modeling (SM) problems wherein the task is to map the observation sequence of agents to the optimal action sequence of agents. Its sequential update scheme is built on the Multi-Agent Advantage Decomposition Theorem Kuba et al. (2021) and Heterogeneous-Agent Proximal Policy Optimization (HAPPO, Kuba et al. (2022)). The lemma provides an intuition guiding the choice of incrementally improving actions, and HAPPO fully leverages the lemma to implement multi-agent trust-region learning with a monotonic improvement guarantee. Unfortunately, HAPPO requests the sequential update scheme in the permutation for agents' orders, meaning that HAPPO cannot be run in parallel. To address the drawback of HAPPO, MAT produces Transformer-based implementation for multi-agent trust-region learning.

Concretely, MAT maintains an encoder-decoder structure where the encoder maps an input sequence of tokens to latent representations. Then the decoder generates a sequence of desired outputs in an auto-regressive manner wherein, at each step of inference, the Transformer takes all previously generated tokens as the input. In other words, MAT treats a team of agents as a sequence, thus implementing the sequence-modeling paradigm for MARL. The encoder $\phi$ takes a sequence of observations $\boldsymbol{o} \triangleq (o^{i_1}, \ldots, o^{i_n})$ in arbitrary order and passes them through $L$ computational blocks. Each of these blocks has a self-attention mechanismVaswani et al. (2017) and a multi-layer perceptron (MLP), as well as residual connections to prevent gradient vanishing and network degradation as depth increases. Thus we can obtain the encoding of the observations as $\hat{\boldsymbol{o}}$ containing interrelationships among agents. Feeding the representations into the value head (an MLP), denoted as $f_\phi$, will get value estimations. The encoder's learning objective is to minimize the individual version of empirical Bellman error by:

$$\mathcal{L}^{\text{MAT}_{encoder}}_{\phi,f_\phi}(\boldsymbol{o}_t) = \frac{1}{Tn} \sum_{m=1}^{n} \sum_{t=0}^{T-1} \left[ R(s, \boldsymbol{a}_t) + \gamma V_{\phi',f'_\phi}\left(\hat{o}^{i_m}_{t+1}\right) - V_{\phi,f_\phi}\left(\hat{o}^{i_m}_t\right) \right]^2 \tag{8}$$

where $\phi'$ is the target network, which is nondifferentiable and updated every few epochs. Meanwhile, the decoder $\theta$ passes the embedding joint action to a sequence of decoding blocks. Crucially, the decoding block replaces the encoder's self-attention mechanism with a masked self-attention mechanism; i.e., the attention of the action to be generated in the current step is computed only among previously computed agents' actions. The output of the last decoder block is a sequence of representations of the joint actions. The same as the value head, this is fed into a policy head (also an MLP), denoted as $f_\theta$, which outputs the policy $\pi^{i_m}_\theta\left(\mathbf{a}^{i_m} \mid \hat{\boldsymbol{o}}^{i_{1:n}}, \boldsymbol{a}^{i_{1:m-1}}\right)$. Besides, the decoder's learning objective is to minimize the clipping PPO objective proposed in HAPPO of

$$\mathcal{L}^{\text{MAT}_{Decoder}}_{\theta,f_\theta}(\boldsymbol{o}_t, \boldsymbol{a}_t) = -\frac{1}{Tn} \sum_{m=1}^{n} \sum_{t=0}^{T-1} \min\left(\mathbf{r}^{i_m}_t(\theta)\hat{A}_t, \text{clip}\left(\mathbf{r}^{i_m}_t(\theta), 1 \pm \epsilon\right)\hat{A}_t\right) \tag{9}$$

where $\mathbf{r}^{i_m}_t(\theta) = \frac{\pi^{i_m}_\theta\left(a^{i_m}_t \mid \hat{\boldsymbol{o}}^{i_{1:n}}_t, \hat{\boldsymbol{a}}^{i_{1:m-1}}_t\right)}{\pi^{i_m}_{\theta_{\text{old}}}\left(a^{i_m}_t \mid \hat{\boldsymbol{o}}^{i_{1:n}}_t, \hat{\boldsymbol{a}}^{i_{1:m-1}}_t\right)}$ and $\hat{A}_t$ is an estimation of the joint advantage function, e.g. GAESchulman et al. (2016).

In practice, the MAT attention mechanism encodes observations and actions using a weight matrix generated by multiplying the embedded queries, as well as keys. The embedded values are multiplied by the weight matrix to output representations. While the encoder's unmasked attention employs a complete weight matrix to extract the interrelationships between agents, the decoder's masked attentions capture sequential actions with triangular matrices. With the properly masked attention mechanism, the decoder can safely output the policy agent-by-agent. We recommend readers give a look at Figure 2, and Algorithm 1 in Wen et al. (2022) to get more details about MAT.

### A.3 MORE DETAILS ABOUT MAQC

Here, we briefly introduce the two scenarios, named *Flock* and *LeaderFollower* in MAQC Panerati et al. (2021).

Denote $i$-th agent's *xyz* coordinates as $\mathbf{x} = (x, y, z)$, individual reward as $r_i$, team reward is $R = \sum_{i=1}^{n} r_i$, and then: (i) In *Flock*, the first agent should keep its position with a predefined location (e.g., $\mathbf{p}$) as close as possible, and $i$-th agents($i > 1$) need to track $(i-1)$-th agent's latitude, i.e., $r_1 = -||\mathbf{p} - \mathbf{x}_1||_2^2$, $r_i = -(y_i - y_{i-1})^2 \ \forall i = 2, \ldots, n$ (ii) The goal of *LeaderFollower* in MAQC is to train the *follower* drones to track the *leader* drone, and the leader drone needs to keep its position with a predefined position as close as possible, i.e., $r_1 = -||\mathbf{p} - \mathbf{x}_1||_2^2$, $r_i = -\frac{1}{n}(z_i - z_1)^2 \ \forall i = 2, \ldots, n$. An overview of MAQC is shown in Figure 10. Note that in our experiments we only use the RGB information provided by the simulator.

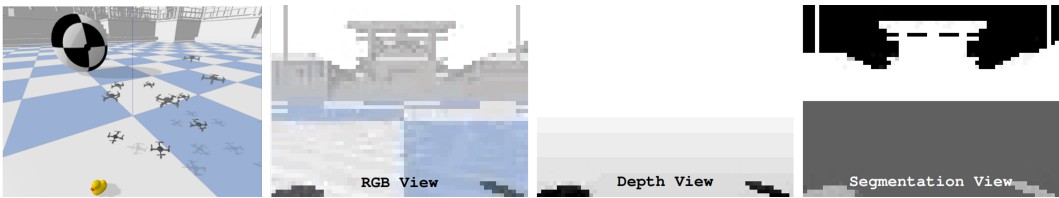

Figure 10: Overview of Multi-Agent Quadcopter Control environment and the observation for the quadrotor. A quadrotor is (i) an easy-to-understand mobile robot platform whose (ii) control can be framed as a continuous states and actions problem but, beyond 1-dimension, (iii) it adds the complexity that many candidate policies lead to unrecoverable states, violating the assumption of the existence of a stationary state distribution on the entailed Markov chain.

## B    Pseudo Code of TIMAR

we list the pseudo code of TIMAR built upon MAT in Algorithm 1

---
**Algorithm 1** Transition-Informed Multi-Agent Representations
---
1: **Input:** Stepsize $\alpha$, number of agents $n$, episodes $K$, steps per episode $T$.
2: **Initialize:** Observation encoder $\phi$, Action decoder $\theta$, Value head $f_\phi$, Policy head $f_\theta$, Replay buffer $\mathcal{B}$, Joint transition model $\hat{\mathcal{T}}$, Online projection head $g$, Online prediction head $q$, Target observation Encoder $\bar{\phi}$, Target projection head $\bar{g}$.
3: **while** Training **do**
4:     Sample a minibatch $(\boldsymbol{o}_t, \boldsymbol{a}_t, r_t, \boldsymbol{o}_{t+1}) \sim \mathcal{B}$.
5:     Calculate $\mathcal{L}_{\phi, f_\phi}^{\text{MAT}_{encoder}}(\boldsymbol{o}_t)$ with Equation (8).
6:     Calculate $\mathcal{L}_{\theta, f_\theta}^{\text{MAT}_{Decoder}}(\hat{\boldsymbol{o}}_t, \boldsymbol{a}_t)$ with Equation (9).
7:     Sample another sequential minibatch $(\boldsymbol{o}_{t:t+K}, \boldsymbol{a}_{t:t+K-1}) \sim \mathcal{B}$.
8:     Get projection/prediction representations $\tilde{\boldsymbol{y}}_{t+1:t+K}$ and $\hat{\boldsymbol{y}}_{t+1:t+K}$ with Equation (5).
9:     Calculate $\mathcal{L}_{\phi, \theta}^{\text{TIMAR}}(\boldsymbol{o}_{t:t+K}, \boldsymbol{a}_{t:t+K-1})$ with Equation (5)
10:    Update the encoder/decoder, value/policy head, joint transition model, and online projection/prediction head by minimizing $\mathcal{L}_{total}$ with gradient descent according to Equation (6).
11:    Update target observation encoder and target projection head with Equation (1)
12: **end while**
---

## C    Full Hyperparameters in TIMAR

We list the full hyperparameters of TIMAR in Table 1.

## D    Hyper-parameter Settings for Experiments

During experiments, the implementations of baseline methods are consistent with their official repositories, all hyper-parameters left unchanged at the origin best-performing status. The hyperparameters adopted for different algorithms and tasks are listed in Table 2-4.

Table 1: Hyperparameters used for TIMAR.

| Hyperparameter | Value |
|---|---|
| Number of prediction steps $K$ | 1 (MA MoJoCo and SMAC) |
| | 2 (MAQC) |
| Auxiliary batch size for TIMAR $B'$ | 128 (MA MoJoCo and MAQC) |
| | 512 (SMAC) |
| Weight for TIMAR loss $\lambda$ | 1 |
| Hidden units in projection/prediction head | 512 |
| Encoder MEA $\tau$ | 0.01 (SMAC) |
| | 0.05 (MA MoJoCo and MAQC) |
| EMA update frequency | 1 |
| Number of blocks for transition model | 1 |
| Number of heads for transition model | 1 |

Table 2: QMIX hyperparameters used for experiments. Parameters with (SMAC) or (SMACv2) after them denote that parameter setting was only used for SMAC or SMACv2 experiments respectively. These are the values in the corresponding configuration file in PyMarl2 (Hu et al., 2021). Mac is the code responsible for marshaling inputs to the neural networks, learner is the code used for learning and the runner determines whether experience is collected in serial or parallel.

| Parameter Name | Value |
|---|---|
| Action Selector | epsilon greedy |
| $\epsilon$ Start | 1.0 |
| $\epsilon$ Finish | 0.05 |
| $\epsilon$ Anneal Time | 100000 |
| Runner | parallel |
| Batch Size Run | 4 |
| Buffer Size | 5000 |
| Batch Size | 128 |
| Optimizer | Adam |
| $t_{\max}$ | 10050000 |
| Target Update Interval | 200 |
| Mac | n_mac |
| Agent | n_rnn |
| Agent Output Type | q |
| Learner | nq_learner |
| Mixer | qmix |
| Mixing Embed Dimension | 32 |
| Hypernet Embed Dimension | 64 |
| Learning Rate | 0.001 |
| $\lambda$ | 0.6 (0.3 for 6h_vs_8z) |

Table 3: Common hyper-parameters used for all methods in the Multi-Agent MuJoCo and Multi-Agent Quadcopter Control domain.

| hyper-parameters | value | hyper-parameters | value | hyper-parameters | value |
|---|---|---|---|---|---|
| gamma | 0.99 | steps | 1e7 | stacked frames | 1 |
| gain | 0.01 | optim eps | 1e-5 | batch size | 4000 |
| training threads | 16 | num mini-batch | 40 | rollout threads | 40 |
| entropy coef | 0.001 | max grad norm | 0.5 | episode length | 100 |
| optimizer | Adam | hidden layer dim | 64 | use huber loss | True |

Table 4: Different hyper-parameter used in the Multi-Agent MuJoCo and Multi-Agent Quadcopter Control domain.

| Maps | TIMAR | MAT | MAPPO | HAPPO |
|---|---|---|---|---|
| critic lr | 5e-5 | 5e-5 | 5e-3 | 5e-3 |
| actor lr | 5e-5 | 5e-5 | 5e-6 | 5e-6 |
| ppo epochs | 10 | 10 | 5 | 5 |
| ppo clip | 0.05 | 0.05 | 0.2 | 0.2 |
| num hidden layer | / | / | 2 | 2 |
| num blocks | 1 | 1 | / | / |
| num head | 1 | 1 | / | / |

