# OpenReview forum: "Boosting Multi-Agent Reinforcement Learning via Transition-Informed Representations"
_ICLR.cc/2024/Conference — Submitted to ICLR 2024_

### Official Review · Reviewer_YZm1 · 2023-10-15

**Soundness:** 2 fair
**Presentation:** 1 poor
**Contribution:** 2 fair
**Rating:** 3
**Confidence:** 4

**Summary:**

This paper proposes to utilize a novel loss term to improve learning in MARL algorithms. Such term is based on a world model used to predict the system's dynamics, and thus the future states. Based on previous literature, it uses an attention mechanism to focus agents on salient information, that they can then use to better coordinate with one another. Empirical results show how the proposed loss term can readily be incorporated into different existing MARL algorithms, and results in improved performances on both image- and state-based problems.

**Strengths:**

The idea of using a world model to predict future information has not been investigated extensively in the MARL setting. In this context, the proposed work is a nice contribution to investigate the applicability of such techniques. The experimental section is wide and well-structured, comparing different baselines on some challenging scenarios.

**Weaknesses:**

I find the work to be really hard to parse and digest: there are many technical details and words that dilute the actual content of the paper itself, making it fairly difficult to get an understanding of what is the true contribution. Please see the Questions below for some more details. Also, the experimental results are good but not amazing, and these small improvements come at a cost of a way more complex computation in general. In there really the need for such a complex machinery just to attain such small benefits? I would have liked to see some experiments where the proposed methodology is really helping (indeed, it seems to do so in the MA-MuJoCo problem), so that to justify this added complexity.

**Questions:**

- The paper is not really clear and easy to follow: it uses a lot of jargon and very long sentences that are just confusing for the reader, while often not really letting the core message of a sentence or paragraph go through. I feel that the paper would greatly benefit from a careful rewriting and polishing process.
- The whole explanation of the attention mechanism and encoding is messy: there are a lot of technical details, but in my opinion what is missing here is a clear and intuitive explanation of what is actually happening. How does the attention model really works? What is it really supposed to do? How can this benefit the learning agents?
- While results on MA-MuJoCo indeed shows the benefits of your proposed approach, those on SMAC (and partly those on MAQC) are not really showing the same kind of improvements. Have you tried with applying your TIMAR optimization to an algorithm other than QMIX in this setting? Or on scenarios where QMIX is in general struggling? Are you able to provide insights on why this is happening?

---

### Official Review · Reviewer_1XWc · 2023-10-24

**Soundness:** 2 fair
**Presentation:** 3 good
**Contribution:** 1 poor
**Rating:** 3
**Confidence:** 4

**Summary:**

This paper introduces a new SSL framework called TIMAR, which utilizes SSL techniques to implicitly incorporate the perception of the global state and observations from other agents by constructing dynamic transition. The experimental results indicate that combining TIMAR with MAT and QMIX, on MA-MuJoCo, SMAC, and MAQC has demonstrated performance improvements over the benchmark MARL algorithms.


Overall, this paper did not provide me with any additional insightful perspectives. It did not attempt to address a MARL problem from a novel angle but rather integrated SSL techniques from single-agent settings into MARL. This improvement is foreseeable, especially in tasks similar to those in MuJoCo.
Furthermore, the performance in tasks outside MuJoCo shows little improvement or is comparable to the baselines. Additionally, the introduction of a highly complex architecture to construct a world model is expensive. Thus I vote to reject the current version.

**Strengths:**

1. The proposed method presents a straightforward and easy-to-follow approach.
2. TIMAR has demonstrated improvements over the baseline in several tasks.

**Weaknesses:**

1. The novelty of this paper is limited. The use of dynamic transition SSL, which has already been proven effective in a single-agent setting, is typically evaluated on tasks in the MUJOCO environment. Therefore, the results presented in MA-MuJoCo in this paper are within the expected range.

2. The introduction of a complex network architecture may potentially lead to significant computational overhead.

3. The experimental section of this paper predominantly showcases TIMAR's advantages in MA-MuJoCo. However, it's worth noting that in such tasks, algorithms like MAPPO are not particularly sample-efficient. In comparison, the original benchmarks proposed in MA-MuJoCo, such as FACMAC, are potentially more efficient due to better utilization of samples. The combination of TIMAR with QMIX in SMAC shows a noticeable performance overlap with the baselines, failing to highlight the algorithm's effectiveness.

**Questions:**

1. Why does TIMAR use different integration approaches with MAT and QMIX? For instance, when combining with MAT, the authors directly apply the loss to the intermediate layer, while with QMIX, an additional layer is added before the RNN. Why not use the same integration approach? Does this additional structure introduce an impact on the results?

2. Could the authors provide more results comparing TIMAR with QMIX and MAPPO on SMAC tasks? It appears that there is a lack of results for MAT and MAPPO on SMAC tasks, and some tasks seem to have not converged.

3. In Figure 7, the tasks involving value loss and Q-values do not align. Additionally, TIMAR seems to exhibit faster initial value loss reduction, intersecting with baselines later. For Q-values, QMIX appears to achieve higher Q-values in the corridor task. Could the authors provide an explanation? This seems to raise questions about the effectiveness of TIMAR. Could the authors elaborate on this?

4. What is the computational overhead associated with TIMAR?

5. In MA-MuJoCo, how are agents configured to perceive information about other agents and the global state?

---

### Official Review · Reviewer_x75r · 2023-10-27

**Soundness:** 2 fair
**Presentation:** 2 fair
**Contribution:** 1 poor
**Rating:** 3
**Confidence:** 4

**Summary:**

This paper proposes a representation learning framework called Transition-Informed Multi-Agent Representations (TIMAR) to improve the performance and sample efficiency of multi-agent reinforcement learning (MARL) methods. The goal of TIMAR is to use an auxiliary objective to learn effective representations in both temporal and agent-level predictive in MARL. The key idea is to train a joint transition model to "implicitly reconstruct the global state and then infer the future observation representation of each agent", which allows better use of agent-level information when learning in partially observable multi-agent tasks.  The proposed framework TIMAR is applied to some existing MARL algorithms (MAT and QMIX) and evaluated in three different multi-agent benchmark environments to demonstrate its effectiveness.

**Strengths:**

- The proposed framework TIMAR is general and can be applied to different existing MARL methods.
- TIMAR is applied to two existing MARL methods and evaluated in three different challenging cooperative MARL benchmark environments.

**Weaknesses:**

- The proposed framework TIMAR looks very similar to the Multi-Agent Masked Attentive Contrastive Learning (MA2CL) framework proposed in [1]. Equations (1), (3), and (6) in the method section look exactly the same with Equations (3), (6), and (9) in [1], respectively. The technical novelty of the paper therefore does not look significant enough to me. Moreover, the similarity between the proposed framework TIMAR and existing work MA2CL should have been highlighted in the introduction of the paper to make the technical contributions clearer. Currently, this does not seem to be mentioned in the introduction at all.
- The experimental results are not that sufficient/convincing in my opinion:
	- Figure 5 shows that, compared to QMIX, the proposed method (after being applied to QMIX) does not really bring any significant final performance improvements in all 4 SMAC maps tested.
	- Section 4.2 is trying to analyse why TIMAR works. However, plotting the value loss and Q-value estimates do not really help provide any intuitive fundamental explanations about why TIMAR works.
	- The generalisation and robustness results presented in Section 4.3 are hard for me to understand. It is not clear to me how the results demonstrate that TIMAR can improve generalisation.
- The clarity of the paper could be significantly improved, especially in terms of how the proposed method works. Also, Section 3.1 should be put in the background or related work section, not in the Our Method section.

[1] Song, Haolin, Mingxiao Feng, Wengang Zhou, and Houqiang Li. "MA2CL: Masked Attentive Contrastive Learning for Multi-Agent Reinforcement Learning." arXiv preprint arXiv:2306.02006 (2023).

**Questions:**

- The only place (I could find) that explicitly explains the key differences between TIMAR and MA2CL is the last sentence in Section 2.3, which says "our method leverages the joint-embedding predictive architecture to learn the surrogate multi-agent world to capture effective knowledge for better multi-agent decision-making". Does this mean the "joint-embedding predictive architecture" is the only novel component in your method? Please clearly explain all the key differences between TIMAR and MA2CL to make the technical contribution clearer.
- What MARL method is MA2CL applied to in the different multi-agent environments in the experiments? MAT and QMIX as in TIMAR? TIMAR and MA2CL should be applied to the same base MARL algorithms when evaluating to ensure a fair comparison. If that is already the case, can you please provide some intuition regarding why TIMAR can bring performance improvement when compared to MA2CL?
- In Figure 8, which learning curve represents TIMAR? It should be coloured as solid grey based on the label but there does not seem to be a grey learning curve. How does this result demonstrate that TIMAR can improve generalisation?

---

### Official Review · Reviewer_vJUQ · 2023-10-31

**Soundness:** 1 poor
**Presentation:** 1 poor
**Contribution:** 2 fair
**Rating:** 5
**Confidence:** 3

**Summary:**

This paper presents a representation learning method for MARL called TIMAR. TIMAR is based on a self-supervised learning loss that encourages the training of encoders that accurately predict the joint observation of all agents across an extended period in the future. This auxiliary loss function is designed to help with (i) inferring the underlying state of an environment given an environment with partial observability and (ii) identifying the effect of each agent's actions on other agents' states and observations. The authors argue that optimizing this auxiliary loss function on top of existing loss functions for MARL training improves the performance and sample efficiency of MARL agents.

The authors tested TIMAR in the SMAC and multiagent MuJoCo environment suites. TIMAR is specifically compared against prior MARL methods such as QMIX, MAPPO, HAPPO, and attention-based methods (MAT). It is argued that the result shows TIMAR outperforming the baselines in terms of expected returns and sample efficiency. Meanwhile, the authors also argued that TIMAR produces better learning representations, as indicated by the MARL methods' lower value loss than the baselines. Further experiments were also done to compare the returns of TIMAR and the baselines when generalizing to slightly different tasks than what they were trained for.

**Strengths:**

**Minor Strength - Quality - Generalization Experiments**

From my perspective, the generalization experiments highlight a strength of the proposed method that can be useful for deep MARL practitioners. However, I believe that the improved generalization capabilities from using TIMAR were not highlighted enough throughout the paper, especially in the introduction section. Furthermore, it would better emphasize this point if the authors provided further analysis of the information contained in TIMAR's learned representation and how it managed to improve generalization performance.

**Major Strength - Significance**

I believe TIMAR can be a promising approach with great significance for MARL (especially related to improving MARL agents' generalization capabilities following the previous point). While the introduced concepts and methods are rather incremental, a lack of self-supervised learning methods for MARL (at least based on my limited knowledge of this topic) makes TIMAR potentially an important paper that can spur further research on representation learning methods for MARL.

**Weaknesses:**

**Minor weakness - Novelty - Incremental Combinations of Previous Works**

The concepts and methods introduced by TIMAR seem to be relatively incremental relative to previous work. As cited in the paper (i.e. regarding BYOL), a similar self-supervised loss function has previously been explored in the single-agent RL case. Furthermore, a similar idea of having an encoder reconstruct the observation of other agents has been explored in the context of agent modelling under partial observability [1], where we control an agent that must collaborate with other potentially unobserved agents with unknown policies. Nonetheless, the combination of these ideas into a MARL method seems to be new, albeit relatively incremental.

**Major weakness - Clarity - Method description**

I had a tough time reading the paper, especially the method description section. Here are a few points that I find difficult to understand:

- **The role of each encoder**. Since the method defines many different encoder networks, it is best if the authors allocate one paragraph to describe the operation within each encoder and their respective roles in TIMAR. Currently, it is not clear how they are working together in TIMAR.

- **Discrepancy between symbols in the method figure and method description**. There are also symbols appearing in the figure (for instance, $\hat{s}_{t}$) that never appeared in the method description. This makes it harder to understand the connection between the provided figure and the method description. This is quite unfortunate, especially since the figure helps readers understand the inner workings of TIMAR.

- **Unclear motivation/details behind several design choices**. Essential details such as (i) why TIMAR's joint transition model implicitly reconstructs the state of the environment (see last few sentences of Section 3.2 point <ii>) and (ii) the motivation behind the choice of using BYOL's self-supervised losses were not provided. Since this is such an important part of the method, I would argue that the authors should have explained this in more detail rather than assuming readers can straightforwardly infer these statements from the writing.

**Major weakness - Clarity and Quality - Experiment Analysis**

In terms of the experiment analysis, I also wish the authors added more details regarding their thought process when analyzing the results. For instance, it could be useful for authors to explain why TIMAR can produce improved performances and sample efficiency in some environments while practically performing similarly in others. Furthermore, I do not follow the reasoning in the representation analysis where authors attempt to explain why TIMAR is better than the baselines based on the magnitude of the value loss. A better analysis would be analyzing the contents of TIMAR's representation to find out if there is important information for decision-making contained in TIMAR's representations that other methods do not manage to capture.

(Minor weakness - Clarity and Quality - Baseline Selection) It is also important to outline the authors' reasoning behind selecting baselines used in this work. It's specifically important for the authors to outline the insights gained by comparing TIMAR to each method.

Citation:

[1] Papoudakis et al. (2021). "Agent Modelling Under Partial Observability for Deep Reinforcement Learning". NeurIPS 2021.

**Questions:**

1. What is the role of each encoder network?
2. Can you describe in detail the operations happening inside each encoder network and how they are integrated in TIMAR?
3. Why do you think TIMAR implicitly reconstructs the environment state?
4. What is the motivation behind using BYOL's self-supervised loss instead of other alternatives?
5. Can you better explain why lower value losses imply better performance when using representations produced by TIMAR?
6. Why was TIMAR outperformed by the baselines in several environments (the first task in quadcopter control and the corridor environment in SMAC)? Similarly, why did TIMAR produce worse generalization performance in the "front_shin" scenario in MuJoCo?
7. What's the reasoning behind selecting some baselines to compare against TIMAR?

---

### Meta-Review · Area_Chair_qMKX · 2023-12-07

**Metareview:**

The reviewers agreed that the paper is not ready to be published due to a lack of novelty and poor quality.
Furthermore, the authors failed to submit a rebuttal.

**Justification For Why Not Higher Score:**

The paper makes marginal contributions and some of the reviewers found it difficult to follow the content due to poor writing.

**Justification For Why Not Lower Score:**

N/A

---

### Decision · Program_Chairs · 2024-01-16

Reject